# Symbiotic Effectiveness, Rhizosphere Competence and Nodule Occupancy of Chickpea Root Nodule Bacteria from Soils in Kununurra Western Australia and Narrabri New South Wales Australia

**DOI:** 10.3390/plants14050809

**Published:** 2025-03-05

**Authors:** Irene Adu Oparah, Rosalind Deaker, Jade Christopher Hartley, Greg Gemell, Elizabeth Hartley, Muhammad Nouman Sohail, Brent Norman Kaiser

**Affiliations:** 1Sydney Institute of Agriculture, School of Life and Environmental Sciences, Faculty of Science, The University of Sydney, Sydney, NSW 2006, Australia; rosalind.deaker@sydney.edu.au (R.D.); jadehartley73@gmail.com (J.C.H.); gemellg6@gmail.com (G.G.); ejohannahartley@gmail.com (E.H.); brent.kaiser@sydney.edu.au (B.N.K.); 2Sydney Institute of Agriculture, School of Life and Environmental Sciences, Faculty of Science, Centre for Carbon, Water and Food, The University of Sydney, 380 Werombi Road, Brownlow Hill, NSW 2570, Australia; nouman.sohail@dpie.nsw.gov.au; 3Elizabeth Macarthur Agricultural Institute New South Wales, Department of Primary Industries and Regional Development, Menangle, Sydney, NSW 2568, Australia

**Keywords:** inoculation, *Mesorhizobium*, nodule occupancy, rhizosphere competence, symbiotic effectiveness

## Abstract

Root nodule bacterial isolates from field-grown chickpea were evaluated in glasshouse and field experiments based on infectivity, relative symbiotic effectiveness, nodule occupancy, plant yield and survivability in the soil rhizosphere for their use as inoculants to enhance chickpea production in Western Australia. Compared to the Australian commercial chickpea inoculant strain *Mesorhizobium ciceri sv. ciceri* CC1192, 10 new strains were ‘fast’ growers, averaging 72 h to grow in culture at 28 °C. The relative symbiotic effectiveness (RSE%) of the new strains in field experiments determined by shoot weight ranged from 77 to 111% in the Desi genotype (var. Kyabra) and 83 to 102% in Kabuli (var. Kimberley Large). Kyabra yielded greater output (2.4–3 t/ha) than Kimberley Large (1.2–1.8 t/ha), with mean 100 seed weights of 23 and 59 g, respectively. The rhizobial strains living in the rhizosphere presented a higher competitive ability for nodule occupancy than those in the bulk soil. Tukey’s multiple comparisons test showed no significant differences between the nodule occupancy ability of the introduced strains (i.e., 3/4, 6/7, N5, N300, K66, K188 and CC1192) in either Kyabra or Kimberley Large (*p* = 0.7321), but the strain competitiveness with each cultivar differed (*p* < 0.0001) for some of the test strains. Strains N5, N300, K72 and 6/7 were the top contenders that matched or beat CC1192 in nitrogen fixation traits. These findings show that new rhizobial strains derived from naturalized soil populations exhibited better adaptability to local soil conditions than CC1192.

## 1. Introduction

Chickpea (*Cicer arietinum*) is an important grain legume grown worldwide for its seed qualities, which include high protein, mineral, and fiber contents [1]. As a legume, it can grow in nitrogen-deprived soils by fixing atmospheric nitrogen through symbiosis with root nodule bacteria called rhizobia [2]. Rhizobia, as bacteroids within the nodule, can fix atmospheric N_2_ to ammonia (NH_3_), a form of N the plant can readily assimilate through the enzymatic activity of glutamine synthetase [3]. Chickpeas were first grown commercially in Australia in the mid-1970s, and prior to that they were grown experimentally [4]. The first rhizobial strain comparison in Australia was undertaken by Corbin et al. [4]. They recommended two inoculant strains, CC1189 and CC1192, with the latter becoming the commercial strain from 1977 [5].

In Australia, chickpeas are mostly sown in winter in medium-to-low-rainfall environments (350–250 mm per annum) [6]. In Western Australia (WA), a significant proportion of pulses are grown in the Ord River Irrigation Area (ORIA) in the far north semi-arid tropical region of the state [6]. The ORIA has an average annual rainfall of 787 mm (compared to Australia’s average of <600 mm) and a wet season spanning from December to the end of March. Approximately 90% of annual rainfall occurs between mid-November and March. The mean maximum temperature is from 31 °C in June to 39 °C in November, while the mean minimum temperature is from 14 °C in July to 25 °C in December [7]. The soil types of the area are Cununurra clay (black soil) and Ord sandy loams (red alluvial soil) [8]. Both soil types have modestly alkaline (pH 7.5–8.0) top-soil and highly alkaline (pH > 8.0) sub-soil [7]. The area sown with chickpea annually in the ORIA ranges between 400 and 1000 ha. Kimberley Large (Kabuli) and Kyabra (Desi) are the most valuable because of their large seeds and high-quality grain, destined for both export and domestic markets [7]. The crop befits the Cununurra clay soil, but also grows well in Ord sandy loams. The yield optimization of Kimberley Large and Kyabra in the ORIA is heavily reliant on inorganic N fertilizer. In some cases, up to 147 kg/ha of N is applied. Stakeholders have little knowledge of how much biologically fixed N is derived from nodulated chickpeas, or whether nodulation is optimal and N-fixation effective. However, after inoculation with an efficacious rhizobial strain, both chickpea types can fix N_2_ effectively for plant growth and thus yield financially lucrative amounts, benefiting the grower, who no longer needs to rely on costly inorganic N fertilizers [9].

Rhizobia are diverse in their capacity to nodulate and fix N_2_ in symbiosis with legume hosts, and there may be diverse groups of indigenous and/or naturalized rhizobia already in the soil that have the capacity to nodulate legumes [5,10]. Some of these rhizobia might have adapted and survived harsh environmental conditions or gained symbiotic genes from introduced strains, making them effective in nodulating the legume [10,11,12,13,14,15]. If a soil contains high numbers of competitive rhizobia that are ineffective in fixing N_2_ with that host, inoculation with an inoculant strain can have a limited effect due to competition, whereby the inoculant strain is outcompeted by the resident soil population for nodule occupancy. Thus, the survival of the inoculant strain on the seed and in the legume–host rhizosphere is crucial to maximize the potential for an effective legume–rhizobia symbiosis. Survival and competitiveness can be the starting point for evaluating the suitability of strains as inoculants [16]. The successful inoculation of the legume relies on the proportion of nodules occupied by the inoculant strain compared to those formed by soil-resident rhizobia [17,18]. However, recent studies have shown that, like the commercial inoculant strains, some resident soil rhizobia can also effectively nodulate chickpea and fix N_2_ efficiently [14,19,20,21,22].

Chickpea root nodule bacteria were initially considered to belong to only two species, *Mesorhizobium ciceri* and *Mesorhizobium mediterraneum* [23]. However, the application of recent molecular techniques has revealed that they belong to several species, including *M*. *amorphae*, *M. loti*, *M. muleiense*, *M. wenxiniae* and *M. haukuii*, within the *Mesorhizobium* genus [24,25,26,27]. The symbiotic genes in *Mesorhizobium* are believed to be located on chromosomal islands or symbiotic integrative conjugative elements (ICESyms) that transfer horizontally between strains, and it has been suggested that this characteristic may be responsible for expanding the host range of *Mesorhizobium* species in chickpea since new species bear *nifH* and *nodC* symbiotic genes similar to *M. ciceri* and *M. mediterraneum* [21,28]. As CC1192 has been the only commercial strain of chickpea *Mesorhizobium* in Australia since 1977, the recent diversity of strains isolated from Australian soils most likely arose from the horizontal gene transfer (HGT) of the ICESym from CC1192, endowing non-nodulating naturalized *Mesorhizobium* strains with the ability to nodulate chickpea [15,21]. Despite recent attention given to plant breeding programs and a resurgence in both applied and basic research into chickpea production in Australia, harvest yields continue to remain low. Current research is directed towards breeding chickpea cultivars with production capacity, while neglecting the important symbiotic role of rhizobia in N_2_-fixation to boost chickpea yield. Untapped genetic potential has been noted in chickpea rhizobia, with research showing diversity in the germplasm for characteristics that can enhance the symbiotic N_2_-fixation capacity and thus increase crop yield [29].

Adverse environmental factors can have a detrimental impact on the quality of the symbiosis between rhizobia and legumes. Therefore, selecting chickpea genotypes and rhizobial strains with the capacity to tolerate environmental stresses could enhance N_2_-fixation and deliver N to the grower, either through elevated seed N and yield or through improved soil N deposition [14,30]. The aim of this study was to compare selected field isolates of chickpea rhizobia against the commercial inoculant strain CC1192 for infectivity, symbiotic effectiveness, rhizosphere competence and competition for nodule occupancy on Kimberley Large (Kabuli) and Kyabra (Desi) chickpea types, in order to identify improved niche rhizobial strains suited to growing conditions in northern Australia and to enhance the N sustainability of chickpea production cropping areas.

## 2. Materials and Methods

### 2.1. Chickpea Seeds

Seeds of Desi (var. Kyabra) and Kabuli (var. Kimberley Large) chickpea selected for their yield and commercial value were provided by Northern Australia Crop Research Alliance (NACRA). Kimberley Large (registration number CV-238, PI 636328) was developed by the germplasm development team at the Centre for Legumes in Mediterranean Agriculture (CLIMA) and the Department of Agriculture Western Australia (DAWA), and released in August 2004 [31]. Its seeds are white and very large, weighing between 55 and 62 g/100 seeds. Comparatively, Kimberley Large is lower-yielding and inconsistent. Kyabra was released in 2005 [32]. It is a high-yielding genotype with good seed quality and excellent early vigor, making it ideal for deep sowing. Its seed weight averages 23 g/100 seeds.

### 2.2. Rhizobial Strains Morphological Characterization of Isolates

Chickpea rhizobial strain *Mesorhizobium ciceri sv. ciceri* CC1192 (current commercial strain) and authenticated field isolates identified as 3/4, 6/7, and F21 were sourced from the SUNFix culture collection held at the University of Sydney, Australia. Strains 3/4 and 6/7 were isolated and authenticated from Desi (var. PBA Slasher) and Kabuli (var. Genesis 090) chickpea plant root nodules, respectively, grown in Narrabri NSW in 2016. These strains outperformed CC1192 and other isolates from the same site in terms of nitrogen fixation in glasshouse trials. Strain 3/4 was the best performer on Desi, while 6/7 was the best performer on Kabuli. Authenticated isolate F21 was isolated from root nodules of Kyabra plants growing in a farmer’s field north of Moree NSW in 2010. It was identified as closely related to *M. metallidurans* STM 2684 and exhibited heat tolerance in laboratory assays (unpublished data). From field trials conducted at the University of Sydney Plant Breeding Institute in Narrabri New South Wales (30.27213° S, 149.80410° E) and the Department of Primary Industries and Regional Development’s (DPIRD) Frank Wise Institute of Tropical Agriculture in Kununurra Western Australia (15.65308° S, 128.70672° E) in 2018, eight additional nodule isolates identified as K8, K66, K72, K188, K203, K208, N5, and N300 were extracted from root nodules on Kyabra and Kimberley Large chickpea plants after 9 weeks of growth (prefix N = Narrabri, K = Kununurra). These isolates were selected from a total of 3200 field isolates based on their infectivity, symbiotic effectiveness, heat tolerance and biochemical characteristics [14]. The 3200 isolates were collected from nine plants each from 80 treatment plots across three random positions in the plots at both Narrabri and Kununurra. Approximately 20 nodules were picked from the harvested plant roots from each plot. Nodules were surface-sterilized with 95% ethanol for 1 min and treated with 5% NaClO for 6 min before rinsing six times with sterile deionized water. Surface-sterilized nodules were individually macerated using a sterile glass rod. The nodule content on the tip of the glass rod was then aseptically streaked across the surface of congo-red yeast mannitol agar (CRYMA) [33]. Streaked plates were incubated at 28 °C for 3–7 days, after which time single colonies showing presumptive rhizobial growth were sub-cultured onto fresh CRYMA medium. After further incubation, each isolate was visually assessed for typical growth characteristics, purity, color and microscopic properties, including Gram stain. Pure cultures of each isolate were stored on yeast mannitol agar (YMA) slopes in 28 mL McCartney bottles [33]. For the long-term storage of pure rhizobial strains, glycerol broth medium was prepared. The medium contained 125.0 g glycerol, 0.5 g yeast extract, 0.5 g NaCl and 1.0 g Lab Lemco powder/1 L in reverse osmosis (RO) water. Then, 1.5 mL of medium was dispensed in 2 mL NUNC cryo-vials and autoclaved at 121 °C for 15 min. After cooling, a substantial amount from each of the pure cultures was looped aseptically into the glycerol medium and kept at −80 °C at the University of Sydney, Australia.

### 2.3. RPO1 RAPD Fingerprinting of the Putative Rhizobial Isolates Bacterial Grouping

For each isolate, rhizobia genomic DNA was extracted directly from a pure broth culture and subjected to Randomly Amplified Polymorphic DNA Polymerase Chain Reaction (RAPD PCR) using random primer oligonucleotide 1 (RPO1) for fingerprinting to identify genetic diversity in bacterial strains, using protocols detailed by Oparah et al. [14].

### 2.4. Authentication of Rhizobial Strains

Seeds of Desi chickpea (Kyabra) were surface-sterilized, pre-geminated, sown singly into each moistened peat pellet and grown under glasshouse conditions [14]. Cultures in yeast mannitol broth (YMB) [33] were prepared from the field-tested inoculant strains CC1192, 3/4, 6/7 and F21 and eight of the DNA fingerprinted nodule isolates, viz. K8, K66, K72, K188, K203, K208, N5, and N300, which showed different RPO1 RAPD fingerprint patterns from strains CC1192, 3/4, 6/7 and F21, and had also demonstrated heat tolerance characteristics. A 200 μL aliquot of each bacterial culture with approximately 1 × 10^9^ cfu per mL was used to separately inoculate 10 replicate Kyabra seedlings [14]. The experiment also included 10 uninoculated replicate plants and 10 replicates with added nitrogen (5 mM KNO_3_) as controls. The inoculated seedlings in the growth assemblies were evenly spaced in a randomized layout within the glasshouse. Six weeks after inoculation, nodule numbers were recorded for each treatment and the shoots were weighed after oven-drying at 60 °C for 72 h.

### 2.5. Symbiotic Effectiveness of Rhizobial Strains

A glasshouse experiment was established at the Centre for Carbon Water and Food (CCWF) in Camden NSW in January 2019 by inoculating 2 chickpea cultivars (Kyabra and Kimberley Large) with the new strains, viz. 3/4, 6/7 and F21, and eight of the DNA fingerprinted nodule isolates, viz. K8, K66, K72, K188, K203, K208, N5, and N300, including CC1192. Replicate custom-designed PVC pots were prepared and connected to a contained gas-exchange system. These were filled with fine sterile gravel and then topped to the brim with approximately 20 cm acid-washed grade-18 silica sand. The growth medium was watered to capacity before pregerminated seeds were sown 1 per pot. For each strain, a loopful of pure bacterial culture was transferred into 1 mL sterile RO water to form a turbid suspension, and then 200 µL of the suspension was used to inoculate surface-sterilized pregerminated seed. The experimental treatments were replicated 6 times and included 6 replicate plants, each of negative rhizobia, and 6 of plus nitrogen (0.5 mM KNO_3_) as controls. Plants were supplied daily with a 1/4 strength nitrate-free Herridge plant growth nutrient solution containing 2500 µm CaCl_2_·2H_2_O, 1000 µm KH_2_PO_4_, 1000 µm K_2_HPO_4_, 100 µm Fe (III)-EDTA, 2000 µm MgSO_4_·7H_2_O, 1500 µm KCl, 11 µm MnCl_2_·2H_2_O, 46 µm H_3_BO_3_, 0.8 µm ZnCl_2_, 0.3 µm CuCl_2_·2H_2_O and Na_2_MoO_4_·2H_2_O/L of sterile water. The treatments were arranged in a randomized complete block design and the glasshouse was set at 26/18 °C day/night temperatures, 60% relative humidity, and a light intensity at pot level of 600 μmol/m^2^ s^−1^ across a 14/10 day/night light period.

At 32 d after sowing, plants in pots were fitted with purposely built lids and sealed with silicone rubber sealants (Qubitac), and then placed under an LED light source (600 μmol m^−2^ s^−1^). The sealed system was connected to an open-flow gas exchange measurement system under N_2_/O_2_ (80/20, *v*/*v*) using electrochemical H_2_ and CO_2_ analyzers (Qubit Systems, Kingston Ontario, Canada). Apparent Nitrogenase Activity (ANA), determined as H_2_ production in air/µmol/h (i.e., H_2_ voltage in air—zero value in air × K × flow rate × volt in air, where K = 0.75 µmol/L) was recorded after attaining a steady H_2_ out-flow from the nodulated root compartment. The N_2_ available to nodules was replaced with Ar by switching the inflow air composition to Ar:O_2_ (80:20, *v*/*v*) so that the electrons that would have been used for N_2_-fixation were channeled to the reduction of protons. Total Nitrogenase Activity (TNA), which was H_2_ production in Ar/µmol/h (i.e., H_2_ voltage in Ar—zero value in Ar × K × flow rate × volt in argon, where K = 1 µmol/L) was then measured non-invasively to observe in real time the variation in activity. Nitrogen Fixation Rate (NFR)/plant was calculated using the equation [34](1)$$NFR/µmol/L=\frac{TNA-ANA}{3}$$
and electron allocation coefficients (EAC), these being the relative allocation of electrons by nitrogenase to H^+^ and N_2_/plant, calculated using the equation [35](2)$$EAC=1-(\frac{ANA}{TNA})$$

The nitrogenase activity/dry nodule weight (i.e., µmol H_2_ g^−1^ nodule dry weight h^−1^) of a measured plant was expressed after harvesting the plant. Nodule numbers, as well as shoot and nodule dry weights per treatment, were recorded. Dry shoot samples were ground into fine constituents and weighed at between 1.25 and 1.35 mg (Sartorius, Göttingen, Germany) into 5 × 9 mm tin capsules (IVA Analysentechnik GmbH & Co. KG, Meerbusch, Germany), and then analyzed for %N on a Delta V Advantage isotope ratio mass spectrometer (IRMS) with a Conflo IV interface (Thermo Fisher Scientific, Bremen, Germany). Following the method of Purcino et al. [36], the relative symbiotic effectiveness percentage (RSE%) of the isolates for N_2_ fixation was calculated using Equation (3) below,(3)$$RSE\left(\%\right)=\frac{Inoculated\;shoot\;dry\;matter}{N\;fertilized\;shoot\;dry\;matter}\times 100\%$$
where RSE% values >80%, 50–80%, 35–50% and <35% were rated as highly effective, effective, poorly effective, and ineffective, respectively.

### 2.6. Field Performance of Selected Strains

#### 2.6.1. Preparation of Inoculants

Peat-based inoculants were prepared for CC1192 and six of the strains, viz. 3/4, 6/7, N5, N300, K66 and K188. These were selected for their unique nodulation, NFR and putative heat tolerance characteristics [14]. For each strain, a culture was grown in yeast mannitol broth (YMB) and incubated at 28 °C until the broth culture contained around 1 × 10^9^ cfu per mL. Inoculants for each strain were prepared by aseptically injecting 100 mL broth culture of each strain singly into 150 g packets of sterile finely milled commercial-grade peat, sealed, and incubated at 28 °C for 14 days to promote the growth and multiplication of rhizobia.

#### 2.6.2. Field Design and Data Collection

A field trial was established at the Department of Primary Industries and Regional Development’s (DPIRD) Frank Wise Institute of Tropical Agriculture in Kununurra Western Australia (15.65308° S, 128.70672° E) in Autumn 2019. The area for irrigation was 0.792 ha and reportedly had no history of inoculated chickpeas for at least four years prior to the establishment of this trial. Prior to planting, a diagonal walk path crossing the whole area was constructed to collect soil samples with a clean trowel every 5–10 m. A total of 20 soil samples were combined and mixed in a large bag. Subsamples of soil mix were tested for indigenous rhizobia by mixing 10 g subsoil in 90 mL of sterile water. Aliquots of 100 µL of each solution were used to inoculate pregerminated Kimberly large seeds. Inoculated plants did not nodulate at 6 weeks after sowing, so it was deemed that the site was free of indigenous rhizobia. The soil type was Cununurra clay with a pH range of 7.5–8.0. Annual average rainfall and maximum/minimum temperatures for the site were 832.7 mm and 35.1/21.2 °C, respectively (Kununurra research facility weather station data), and available soil N averaged 30 kg N/ha, to a depth of 30 cm.

The seven strains, viz. 3/4, 6/7, N5, N300, K66, K188 and CC1192, were prepared as peat slurries at a rate of 250 g peat suspended in 500 mL sterile water, sufficient to inoculate 100 kg chickpea seeds. Seeds of Kyabra and Kimberley Large were then slurry-inoculated at the recommended rate and sown into irrigated raised beds of length 11 m and size 19.8 m^2^. Nitrogen-fed (60 kg N/ha) and uninoculated plants were included as controls to make a total of nine treatments. There were four replicate plots per treatment, and the seeding rates for Kimberley Large and Kyabra were 113 and 44 kg/ha, respectively. The estimated germination rate and plants/m^2^ values were 85% and 15 plants/m^2^, respectively. Each strain was restricted to one row, and there was one buffer row between treatment plots. The treatments were arranged in a randomized complete block design.

Data collection was performed at weeks six and nine and at final grain harvest. At six weeks after sowing, nine plants were dug from each treatment plot across three random positions within the plot to visually assess nodulation. At nine weeks, nodule dry weight and shoot dry matter were measured after drying for 72 h at 60 °C. Grain yield and 100 seed weights were estimated at plant maturity five months after sowing. Dry shoot and seed samples were ground into fine constituents (SPEX Sample Prep 2010, Geno/Grinder, Metuchen, NJ, USA), weighed between 30 and 40 mg (Sartorius), placed into ultra-light-weight 37 × 37 mm tin foil squares and encapsulated to a 9 mm pellet diameter (Manual press 9 mm pellet diameter 41.01-0004), and were then analyzed for %N on a CHNS analyzer vario MACRO cube (Elementar Analysensysteme GmbH, Langenselbold, Germany) using Equation (4), shown below [34].(4)$$Plant\;N\left(g\right)=\frac{SDW\left(g\right)\times \%N}{100}\times 1000$$

The relative symbiotic effectiveness (RSE%) of the test strains was calculated by comparing the shoot dry weight of each test strain treatment with that of CC1192 as the positive reference control [14,22,37,38], using the equation $RSE\;\%=\frac{X}{y}\times 100$, where x is the shoot dry weight of plants inoculated with test strains and y is the shoot dry weight of CC1192-inoculated plants. The RSE% of strains greater than 80%, ranging from 60 to 80% and less than 60%, were rated as effective, poorly effective, and ineffective, respectively.

### 2.7. Rhizosphere Competence and Nodule Occupancy of Rhizobial Strains

The most probable number (MPN) of viable inoculant rhizobia in root rhizospheres was estimated from rhizosheath soil washed and collected from the roots of plants sampled at nine weeks [39]. The RAPD profiles of background strains isolated from nodules collected from roots of plants inoculated with root wash suspension were compared with the introduced inoculant strains to determine the percentage of introduced rhizobial populations against the background population. Roots of nine plants per plot collected from the field were washed in 100 mL sterile water to form a suspension of rhizosphere constituents. A ten-fold dilution series was prepared from each suspension, and 100 µL aliquots from each of the 10^−0^, 10^−1^, 10^−2^, 10^−3^ and 10^−4^ dilutions were used to inoculate three plants per dilution level growing in Gemell growth assemblies (EJ Hartley and LG Gemell, personal communication). Inoculated seeds were placed on upturned test tubes, which had been wrapped in a non-bleached paper towel, and placed in 700 mL plastic cups containing 400 mL sterile nitrate-free Jensen’s plant growth nutrient solution. Each cup was covered with a dome lid and the plants were grown under similar conditions as those described in the rhizobia authentication experiment above. Added N (0.1% KNO_3_) and uninoculated controls were included to confirm growth potential and no cross-contamination, respectively. Rhizosphere samples from field-grown Kimberley Large plants were applied to Kimberley Large seeds, and samples from field-grown Kyabra plants were applied to Kyabra seeds in the growth assemblies. Nodulation was assessed six weeks after sowing. The MPN of viable rhizobia in the rhizosphere soil for each treatment was determined using a probability table [40].

In addition, isolates from 1600 nodules sampled from Kununurra nine weeks after sowing were analyzed using RPO1 RAPD fingerprinting to determine the nodule occupancy abilities of the inoculant strains [41]. Based on the fingerprint patterns, the isolates were identified as either an inoculant strain or from a background soil rhizobia population. The fingerprint patterns of nodule isolates were compared to the fingerprints of their source plot inoculant strains (i.e., 3/4, 6/7, N5, N300, K66, K188 and CC1192) to estimate the ratio of nodule occupancy of inoculant strains versus soil resident population.

### 2.8. Statistical Analysis

All statistical analyses were performed using GraphPad Prism version 9.1.2 for Windows (GraphPad Software, San Diego, CA, USA) and Microsoft excel. Analysis of variance (ANOVA) was used to test for the interactive and non-interactive effects of all treatments in the glasshouse and field experiments. Mean comparisons of grain yield and 100 seed weight were performed when analysis of variance (AOV) treatment P(F) values were significant, and where there were missing data, the most effective replicates used for mean comparisons were adjusted.

## 3. Results

### 3.1. Morphological Characteristics of Isolates

There was diversity in the growth rates among the isolates screened on CRYMA medium, with some producing uniform colonies and others mixed. The colony morphology was circular, creamy pink, smooth and raised, with an entire margin and mostly 1–3 mm in diameter. Compared to CC1192, which takes 5 days to grow at 28 °C, the isolates grew faster and began to appear after 3 days. The isolates were Gram-negative and rod-like in shape, exhibiting typical characteristics for rhizobia [33].

### 3.2. RPO1 RAPD Fingerprints of the Putative Rhizobial Isolates

There was diversity in the genetic fingerprints of the isolates, and many were genetically distinct from the original inoculant strains applied in the 2018 field trial. Red lines highlight similar bands in some of the isolates, which could indicate shared genetic material between isolates. An example is shown in Figure 1, Gel 1. A similar matching of the bands can also be observed across some of the isolates in Gel 2. Approximately 15% of the isolates from Kununurra resembled the inoculant strains CC1192, 3/4 and 6/7 in their fingerprints. According to a statistical Mantel test, the diversity among strains did not correlate to their soil sampling site (*p* ≤ 0.05). RAPD-PCR genomic fingerprint patterns of the newly selected strains showed differences from *M. ciceri sv. ciceri* CC1192 (Figure 1, Gel 2).

### 3.3. Authentication of Rhizobial Strains

The eight isolates selected for authentication based on their unique RPO1 fingerprints nodulated Kyabra plants along with reference strains 3/4, 6/7 F21 and CC1192. However, some of the treatments (i.e., the different letters on bars) varied significantly in nodulation (*p* = 0.0001) and dry shoot weight (*p* < 0.0001) (Figure 2a,b). After 6 weeks of sowing, chickpea plants inoculated with strain 3/4 presented the highest nodule number per plant (21 nodules) and an average dry shoot weight of 8 g per plant. Based on dry shoot weight, strains 3/4, 6/7, N300 and N5 were significantly more effective than CC1192. Of the remaining strains, K188, K66 and K72 were near equal to CC1192, while F21 performed poorly. The uninoculated control plants did not nodulate and had the lowest dry shoot weight (Figure 2b).

### 3.4. Symbiosis Effectiveness of Rhizobial Strains

Data from the glasshouse experiment enabled a detailed examination of rhizobial effectiveness using the parameters of nodulation, nitrogen fixation (i.e., undisturbed H_2_-evolution assay—NFR) and nitrogen allocation to aerial tissues (total plant N). The results show that, across the two chickpea lines and rhizobial treatment combinations, the key plant traits of nodule number, nodule development, shoot growth, plant N and steady-state Nitrogen Fixation Rates varied (Table 1). At 32 d, inoculated plants of both cultivars were nodulated.

For Kyabra, the nodule number per plant ranged from 44 (strain K66) to 78 (strain 6/7). Nodule dry weights across inoculated treatments varied from 0.057 to 0.122 g. Strain K66 gave a significantly lower value the other strains. Shoot dry weights were not significantly different (*p* > 0.05). Strain K188 produced the greatest weight of 0.527 g, which was slightly more than the N-fed plants, while the uninoculated plants gave the lowest yield, at 0.291 g. The nitrogen fixation rate ranged from 32.4 (strain K8) to 51.6 (strain N5) µmol H_2_ g^−^ DW nodule per plant. Strain N5 was 30% higher in NFR than CC1192. Three strains, K72, N5 and K188, produced the highest plant nitrogen content, with values of 25, 22.3 and 20.5 mg plant N per plant, respectively. These data were 46, 30 and 20% higher in PN than CC1192, respectively. Symbiotic effectiveness ranged from 60.7% (K66) to 102.9% (K188).

For Kimberley Large, the nodule number per plant ranged from 53 (strain K188) to 88 (strain 3/4).

Nodule dry weights across inoculated treatments varied from 0.065 to 0.116 g, and were not significantly different (*p* > 0.05). Shoot dry weights were not significantly different (*p* > 0.05). Strain 6/7 produced the highest weight of 0.751 g, a small increase over the N-fed plants, while the uninoculated plants were lowest, yielding 0.247 g. The nitrogen fixation rate ranged from 36.4 (strain K203) to 53.0 (strain N5) µmol H_2_ g^−^ DW nodule per plant. Strain N5 was 16% higher in NFR than CC1192. Three strains—K72, 6/7 and K66—produced the highest plant nitrogen contents, with values of 29.8, 28.1 and 26.4 mg plant N per plant, respectively. These values were significantly (106, 90 and 82%) higher (*p* < 0.05) in PN than in CC1192, respectively. Symbiotic effectiveness ranged from 69.3 (K66) to 114.8 (6/7).

### 3.5. Field Performance of Selected Strains

Rhizobial strain, plant genotype and plant genotype × rhizobial strain interaction made significant contributions to the total sources of variation between treatments in nodule DW and shoot DW (Table 2). The effects of strain, plant genotype and the interaction of strain and plant genotype contributed 23, 0.2 and 26%, respectively, to RSE%, and while plant genotype had no significant effect (*p* = 0.676) on RSE%, strain (*p* = 0.0076) and strain × plant genotype (*p* = 0.0035) did.

At the Kununurra field trial, the symbiotic effectiveness and N_2_-fixation abilities of strains 3/4, 6/7, N5, N300, K66 and K188 were compared with the commercial strain CC1192. Plant growth in the first harvest period after six weeks was found to be similar across all plant/rhizobia combinations, as well as the uninoculated and +N treatments. At this early stage, nodulation was minimal to non-existent in both Kyabra and Kimberley Large (Figure 3). At nine weeks of growth, nodulation and nodule growth increased significantly in both chickpea varieties, with large groups of indeterminate nodules often fused together present on the roots. The fused nodules compromised the task of recording individual nodule counts, so a net nodule dry weight (DW) measurement was used to determine the nodulation potential of each strain. Nodule DW was generally lower and more variable for Kimberley Large than Kyabra (Figure 4a). Both the uninoculated and +N treatments showed significantly lower nodule DW. RSE% ranged from 77 to 111% for cv. Kyabra and 83 to 102% for cv. Kimberley Large (Table 3). In both Kyabra and Kimberley Large treatments, K66-inoculated plants had the highest RSE% (i.e., 111 and 102%, respectively), and these values were slightly higher than the CC1192 treatment. There was no significant difference (*p* > 0.05) between the shoot DW values of all the test strains and the CC1192 treatment in the Kimberley Large treatment (Figure 4b). For Kyabra, strains CC1192, 3/4, K66 and K188 produced the highest total N (g/plant) compared to the remaining treatments, with CC1192 yielding the most (Figure 4c). For Kimberley Large, CC192 and 3/4 were again amongst the four best strains that produced the highest total N, along with two other strains, 6/7 and N5. Strain N5 yielded the highest total N/plant (Figure 4c).

In general, the grain yield of Kyabra was higher than that of Kimberley Large, as shown in Table 4. However, none of the test strains outperformed the commercial strain CC1192 in terms of final yield. The grain yield of plants inoculated with the test strains averaged between 7 and 10% less than CC1192 and the +N treatments, respectively, but 11% more than the uninoculated. In Kimberley Large, strain 3/4-inoculated plants had the highest grain yield (equivalent to 1.7 t/ha) among the test strains. However, this was nearly 17% less than the +N treatment, but 10 and 17% more than CC1192 and the uninoculated treatments, respectively. Plant genotype, rhizobia strain and plant genotype × rhizobial strain contributed 79, 7 and 1.6%, respectively, to the total source of variation in grain yield. The mean 100 seed weights of Kyabra and Kimberley Large were 23 and 59 g, respectively, and nodulation did not predict grain yield. None of the rhizobia treatments in either Kyabra or Kimberley Large differed significantly (*p* > 0.05) from CC1192 inoculation in terms of 100 seed weight and seed %N content.

### 3.6. Rhizosphere Competence and Nodule Occupancy of Rhizobial Strains

All selected novel strains formed nodules on their respective host chickpea types (Figure 5a). The MPNs recorded for each treatment indicate that inoculant strains survived and multiplied in the root rhizosphere. The MPN of rhizobia-nodulating Kimberley Large was low compared to Kyabra for all strains. Soils from the uninoculated and +N plots had lower MPNs of rhizobia compared to the inoculated plots, indicating the rhizosphere was dominated by inoculant strains over resident soil rhizobia, especially in the Kyabra rhizosphere (Figure 5a). A proportion of the nodules were occupied by naturalized soil rhizobia populations (Figure 5b). However, in all treatments, the inoculant strains had the highest proportion of nodule occupancy. In Kyabra, strains 3/4 and N5 had the highest nodule occupancy (almost 90%), which was approximately 10% more than in the CC1192 treatment, while in Kimberley Large, strains 3/4, N5, K66 and K188 had the highest (almost 90% each), which were about 5% more than CC1192. While there was variation in nodule occupancy, Tukey’s multiple comparisons tests showed no significant difference in nodule occupancy between Kyabra or Kimberley Large (*p* = 0.7321). However, the nodule occupancy by inoculant strains varied between each cultivar for some of the test strains (*p* < 0.0001).

## 4. Discussion

We have provided further evidence of the presence of a naturalized population of diverse chickpea rhizobia in some Australian soils. Despite the introduction of a single inoculant strain for chickpea in Australia, several new root nodule bacteria were isolated from field-grown Desi and Kabuli chickpeas in Kununurra and Narrabri. This is consistent with recent studies on chickpea *Mesorhizobium* diversity in Australian soils [22,38]. The new strains exhibited variable nodulation and symbiotic effectiveness in glasshouse trials, with some performing better than the current inoculant strain; however, there was little variation in symbiotic performance and grain yield when applied to chickpea in the field in Kununurra. The lack of variation in field performance is also consistent with the previous findings of Elias and Herridge [38] and Zaw et al. [22], and may be explained by the origin of diversity stemming from the horizontal gene transfer of the integrative conjugative element from CC1192 (ICE*Mc*Sym^1192^) to resident *Mesorhizobium* in the soil. Rathjen et al. [42] reported that newly isolated strains from soils in Australia and Myanmar, when inoculated onto chickpea, did not outperform the current Australian commercial strain *Mesorhizobium ciceri* CC1192. Myanmar strains exhibited the poor nodulation of chickpea plants when tested under Australian field conditions.

In addition, where plants are grown in ideal glasshouse conditions using N-free nutrients, it is easier to measure rhizobia quality and effectiveness on the plants, whereas the complexity and criticality of natural environmental conditions (such as soil temperature, moisture and pH) could influence symbiotic relationships and plant growth. Also, the interaction of plants with pests and pathogens, as well as predation from other soil organism, could lower live rhizobia numbers upon introduction into the soil. Moreover, the N content of the field soil may not have been low enough to allow significant differences in strains’ infectivity and symbiotic effectiveness to be detected. Kimberly Large, for example, usually displays a strong preference for exogenous N in its growth and yield, and shows significant nodulation sensitivity to elevated N in the field experiments. Corbin et al. [4] reported that soil nitrate obscured differences between strains.

### 4.1. Diversity of Chickpea Nodulating Rhizobia from Kununurra and Narrabri

Compared to CC1192, most isolates differed in colony morphology, size, and time to grow on CRYMA. A few isolates closely resembled CC1192, but many others grow faster than CC1192. The morphological appearance, the presence of a Gram-negative strain showing rod-like shaped cells, and the lack of Congo red absorption by colonies (i.e., pink colonies, not red) when grown on CRYMA suggest the isolates are typical rhizobia [43]. Previous researchers also characterized *Mesorhizobium* isolates on the basis of colony shape, color, texture and growth, and reported that *Mesorhizobium* was Gram-negative, motile and rod-shaped, and that both fast and slow growers exist [41,44]. A study by Küçük and Kıvanç [45] characterizing chickpea nodule isolates reported that chickpea rhizobia were Gram-negative and rod-like in shape, and had moderate motility. The clear variation in the RAPD-PCR fingerprints among the isolates evidenced that the nodules sampled in 2018 from two fields contained various chickpea rhizobial populations distinct from the introduced strains CC1192, 3/4, 6/7 and F21. The minimal recovery of the introduced strains in nodules sampled from Kununurra suggests that most of the inoculant strains did not survive the harsh environment, or were outcompeted by the native rhizobia.

The infectivity and effectiveness of the eight new strains varied when inoculated onto Kyabra chickpea under N-free growth conditions in the glasshouse, as indicated by the variation in nodulation and shoot biomass. Since none of inoculated chickpea showed visible signs of N deficiency and often produced greater biomass than the uninoculated plants, it could be suggested that the strains nodulated and fixed N_2_ adequately. Previous studies have reported variation in the N_2_-fixation capabilities of chickpea rhizobia. Zafar et al. [20] isolated 17 strains from chickpea growing in Pakistan and the USA, out of which 10 exhibited N_2_ fixation activity but at different levels. Zaw et al. [22] observed some relationship between chickpea cropping history and the SE % of isolated strains, with the highest SE % seen in chickpea cropped soils and the lowest in soils where chickpea had not been grown for more than 10 years. Furthermore, Elias and Herridge [19] reported a variation in the infectivity and symbiotic effectiveness of different chickpea nodulating rhizobia isolated from cropping soils in northern NSW Australia.

### 4.2. N_2_ Fixation Traits in Whole Plant in the H_2_ Evolution Assay

Inoculation with rhizobia increased aboveground biomass compared to the uninoculated treatments. Plants inoculated by some of the new strains had higher dry shoot weights than CC1192-inoculated plants. High nodulation in some of the inoculated host/strain treatments including Kyabra × N300, Kimberley Large × 3/4 and Kimberley Large × CC1192 did not lead to increased nodule mass. It has been suggested that increased nodule number may not result in increased nodule mass because of a possible trade-off between the two parameters (i.e., inoculation with a strain may cause the host plant to produce many nodules, but the nodules may be smaller in size and weight) [46].

A lower N_2_ fixation rate corresponding with increased dry shoot weight in some host/strain treatments (e.g., Kyabra × K188, Kyabra × K72 and Kimberley Large × 6/7) could imply less carbon demand by some bacteroids from their symbiotic partners, while a high N_2_ fixation rate corresponding with a comparatively low dry shoot weight (e.g., Kyabra × N5, Kyabra × 3/4 and Kyabra × K66) suggests carbon allocation to bacteroids could have been greater [47]. Rhizobia can vary greatly in their ability to fix N and or the amount of C reward demanded from their host. As nodules mature and begin to fix N, the movements of N and C metabolites between the plant and the matured nodules impact the whole of plant growth, and therefore the nutritional demand can cause plants to activate N signaling processes that regulate symbiotic interaction and partner selection [48]. Thus, variations among a strain’s N_2_-fixation capacity could be partly attributed to differences in the bacteroid C use efficiency, which could be why a strain like K72 was of lower quality when in partnership with Kimberley Large in terms of its N_2_ fixation ability, but it was of higher quality with Kyabra.

### 4.3. Symbiotic Performance of Selected Nodule Isolates in the Field at Kununurra

Inoculation with all new strains increased nodulation in both Kyabra and Kimberley Large chickpea varieties at Kununurra. The nodulation of plants in the uninoculated treatments indicates the presence of naturalized rhizobia in the soil, and the diversity of these isolates was confirmed by RPO1 RAPD fingerprinting. Although nodules sampled from the uninoculated plots were pink inside, showing a leghemoglobin function, the naturalized population in the soil may not have been sufficient to outcompete the inoculant strain for root infection, which could explain why there was higher nodulation in the inoculated plots than in the uninoculated plots. Up to 2 to 5 weeks after sowing, when active biological N_2_ fixation may have begun, the plants utilized seed and available soil N, since there was no significant variation in traits measured at 6 weeks after sowing [49]. The inoculation with introduced strains enhanced nodulation. The nodule occupancy data show that there were variations in the infectivity capacities of the new strains, with some higher, equal to, or less than that of CC1192.

There were significant differences in total N between treatments. C and N metabolites supplied by the plant fuel nodule activities and development, hence inoculated plants that formed more nodules but were low in plant N compared to other treatments may have encountered high-cost symbiosis, where the increased partitioning of nutrients to maintain activity may have happened at the expense of plant vegetative growth [50]. Although some of the rhizobia treatments increased grain yield compared to the uninoculated treatment, the increments were insignificant, which suggests that either the naturalized soil rhizobia were effective in fixing N_2_, or that soil available N was sufficient to meet the plants’ N demand. Since the amount of N administered in the +N treatment may not have been enough to repress nodulation, the higher grain yield recorded could be due to the combined effects of exogenous N and N_2_ fixed by naturalized soil rhizobia.

### 4.4. Rhizosphere Competence and Nodule Occupancy of Selected Rhizobial Strains

There was a clear benefit of inoculation with selected strains in the nodulation of both Kyabra and Kimberley Large chickpea plants. The capacity of a strain to nodulate a legume host and effectively fix N_2_ depends partly on whether the strain survives during inoculation and in the rhizosphere to outcompete indigenous soil rhizobia for high nodule occupancy [10,51]. It is vital that inoculant strains survive challenging conditions in the target environment, such as predation by other soil organisms and competition from naturalized rhizobia that may have already adapted to the environment [38,52]. Hence, to evaluate whether the selected strains may be candidates for commercial inoculant production, their rhizosphere competence in target soils and nodule occupancy were evaluated.

The nodule occupancy data illustrate variations in the infectivity of the introduced inoculant strains, with some higher, equal to, or less than that of CC1192. Like CC1192, each of the newly selected strains had a significantly higher nodule occupancy than the naturalized soil strains, suggesting that they were more competitive than the naturalized populations because of the high numbers of inoculants applied at sowing [53]. It is possible that, in some instances, certain resident rhizobia could have the same RAPD profile as the inoculant strain, thereby inflating the inoculant ‘number’. Moreover, the N status of the plant affects nodule numbers, but not [14] necessarily competitiveness between strains in forming nodules. Thus, the plants may have been unable to discriminate between ineffective and effective rhizobia due to the high N availability prior to the onset of N_2_ fixation [50]. The lower MPNs of nodulating rhizobia in the rhizosphere of Kimberley Large compared with Kyabra at six weeks after sowing may explain the delayed nodulation and lower nodule dry weight in this chickpea variety. There was no relationship between the geographical location from which the strains were isolated and nodule occupancy, with no significant difference between strains from Narrabri and Kununurra. This indicates that the adaptation of rhizobia to the plant host environment may be more important than its adaptation to local edaphic and climatic conditions. In a previous study [14], we found that amongst the new strains included in this study, there was diversity in terms of abiotic stress tolerance, but that diversity did not correlate with the strains’ respective sources of origin.

## 5. Conclusions

This study has identified new chickpea rhizobia that exhibit genotypic, phenotypic and morphological diversity, highlighting relevant variations that should be considered when selecting new strains for commercial use. The nodulation of Desi and Kabuli chickpea varieties was improved after inoculation with selected rhizobial strains in the field, indicating superior growth and survival compared with chickpea nodulated by resident soil rhizobia. However, while differences between strains in symbiotic effectiveness were more pronounced in controlled glasshouse experiments, there was less difference exhibited in the field. Strains N5, N300, K72 and 6/7 can represent suitable inoculant candidates for the ORIA, as they were effective when applied to both Desi and Kabuli chickpea varietal types. Also, strain N5 performed better than the current commercial strain CC1192 for both Desi and Kabuli chickpeas. It is also worth noting that not only did strain N5 display a high MPN and nodule occupancy ability, but it was also previously demonstrated to have strong heat tolerance [14]. It is essential to continue the field screening of these new strains to identify those that can manage specific environments, particularly those that induce abiotic stresses, such as the high temperatures in the ORIA. Since N can impact both nodulation and N_2_ fixation, soil N needs to be managed to allow the degree of N_2_ fixation to be maximized in chickpea, thereby reducing the use of costly inorganic nitrogen fertilizer. These findings demonstrate that rhizobial strains can be sourced from naturalized soil populations with improved symbiotic efficiency, and that isolates from some Australian soils may be better adapted to local soil conditions than CC1192, which would confer advantages in terms of survival, nodulation and nitrogen fixation in chickpea.

## Figures and Tables

**Figure 1 plants-14-00809-f001:**
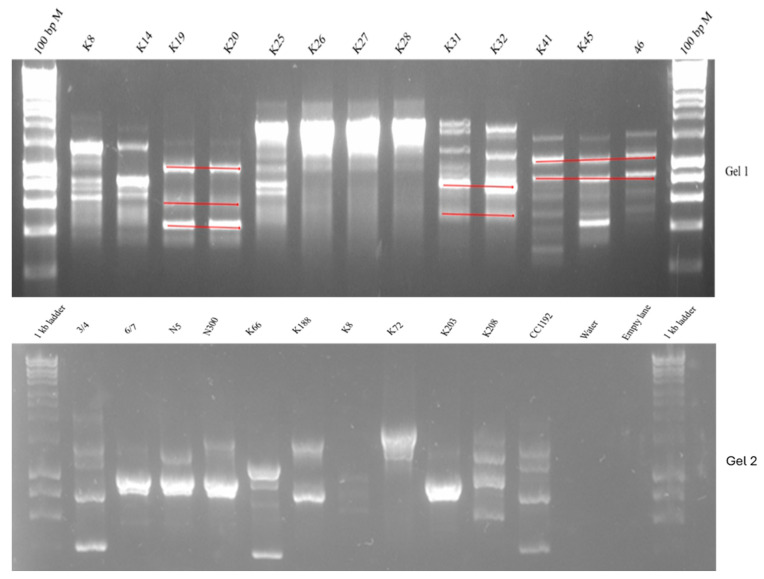
RAPD-PCR fingerprint pattern generated with the RPO1 primer showing diversity in novel chickpea rhizobia strains, *M. ciceri sv. ciceri* CC1192 and negative water control. Strain prefix N and K were sourced from Narrabri and Kununurra, respectively. The first and last lane of Gel 1 contained a Bioline HyperLadder^TM^ 100 bp (Bio-33029) molecular marker, while in Gel 2, the molecular marker was Bioline HyperLadder^TM^ 1 kb (Bio-33053). In Gel 2, the lane between water control and the ladder was not used. In RAPD-PCR fingerprinting, similar bands like those marked with red lines on Gel 1 represent same-sized DNA fragments. The banding patterns of some isolates differed from those of the commercial inoculant strain CC1192. However, as seen in Gel 2, there is a close relationship between CC1192 and strain ¾. Strains 6/7, N5, and N300, which originated from Kabuli plants in Narrabri, had similar bands.

**Figure 2 plants-14-00809-f002:**
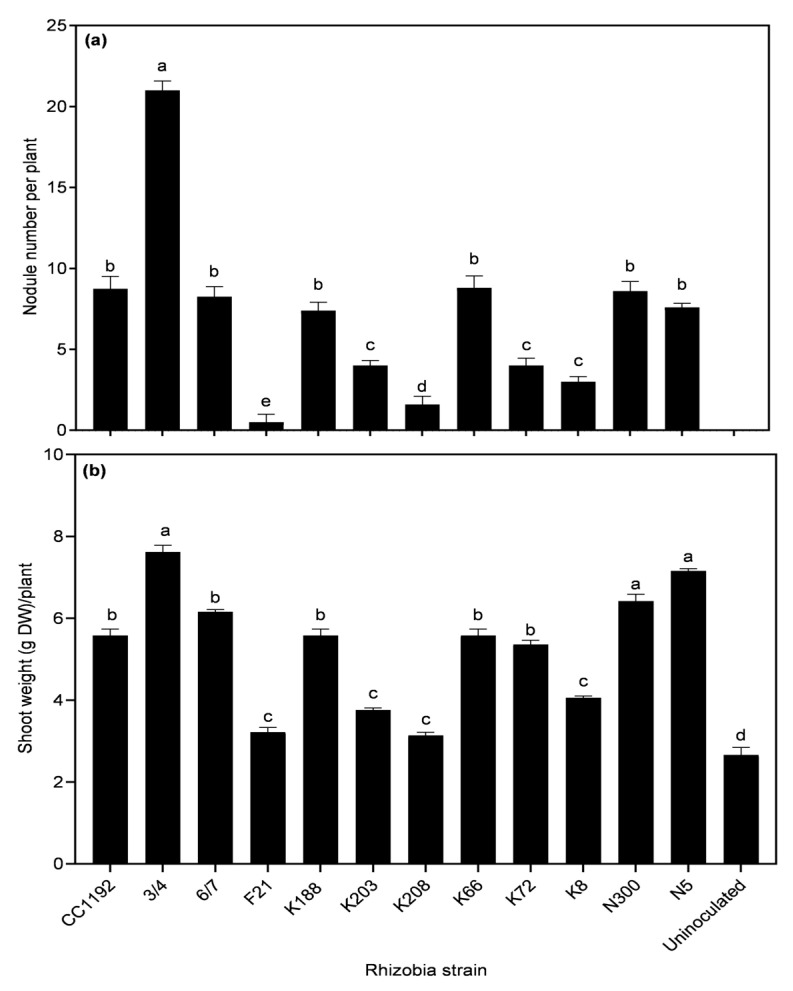
Relationship between (**a**) nodule number and (**b**) shoot DW per plant of Kyabra plants inoculated with eight diverse chickpea root nodule isolates and strain 3/4, 6/7 and F21 at six weeks after sowing. CC1192 was included as a reference. Data represented as mean ± SEM (*n* = 10 plants per treatment). Treatments that share the same letter are not significantly different according to Dunnett’s multiple comparison test (*p* ≤ 0.05).

**Figure 3 plants-14-00809-f003:**
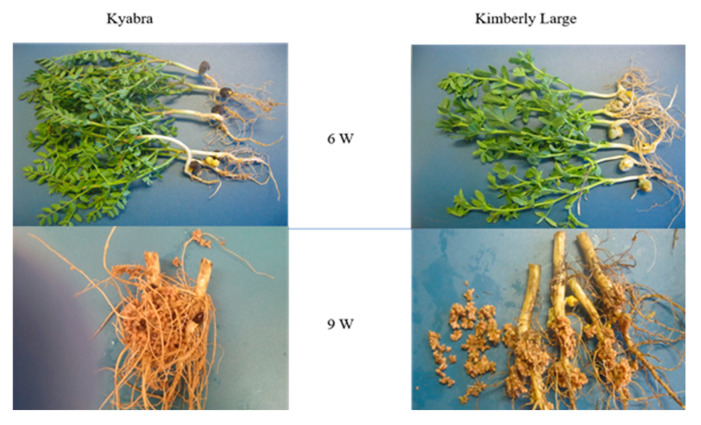
Representative plant appearance and nodulation characteristics in 6- and 9-week field-grown Kyabra and Kimberly Large chickpea plants inoculated with strain N5.

**Figure 4 plants-14-00809-f004:**
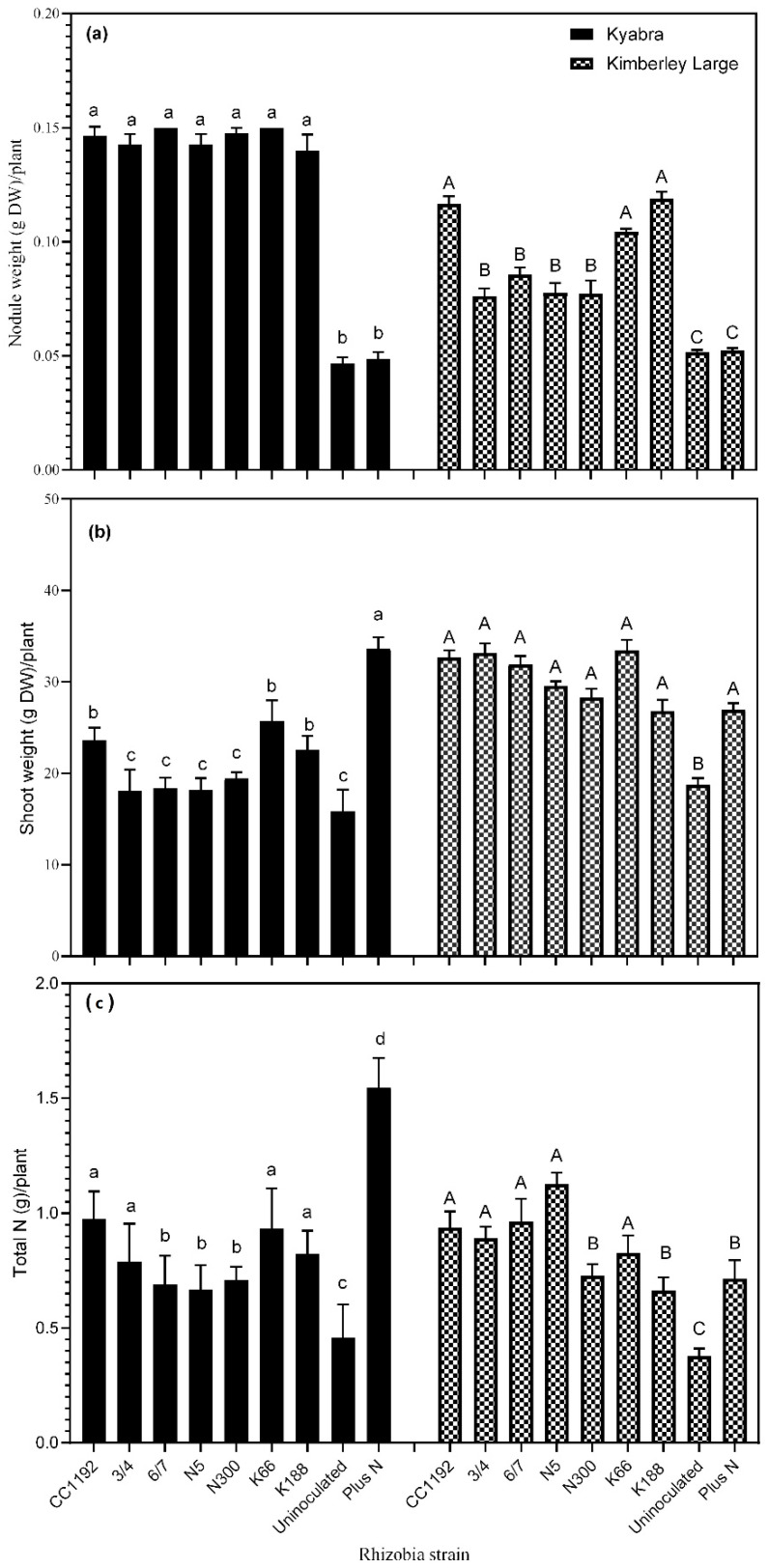
Relationship between strain and (**a**) nodule DW, (**b**) shoot DW and (**c**) plant N at nine weeks in Kyabra and Kimberley Large in the 2019 Kununurra field experiment. Data represent mean ± SEM (*n* = four plots consisting of nine plants per plot). Treatments that share the same letter (lowercase for Kyabra bars and uppercase for Kimberley Large) are not significantly different according to Dunnett’s multiple comparison test (*p* ≤ 0.05).

**Figure 5 plants-14-00809-f005:**
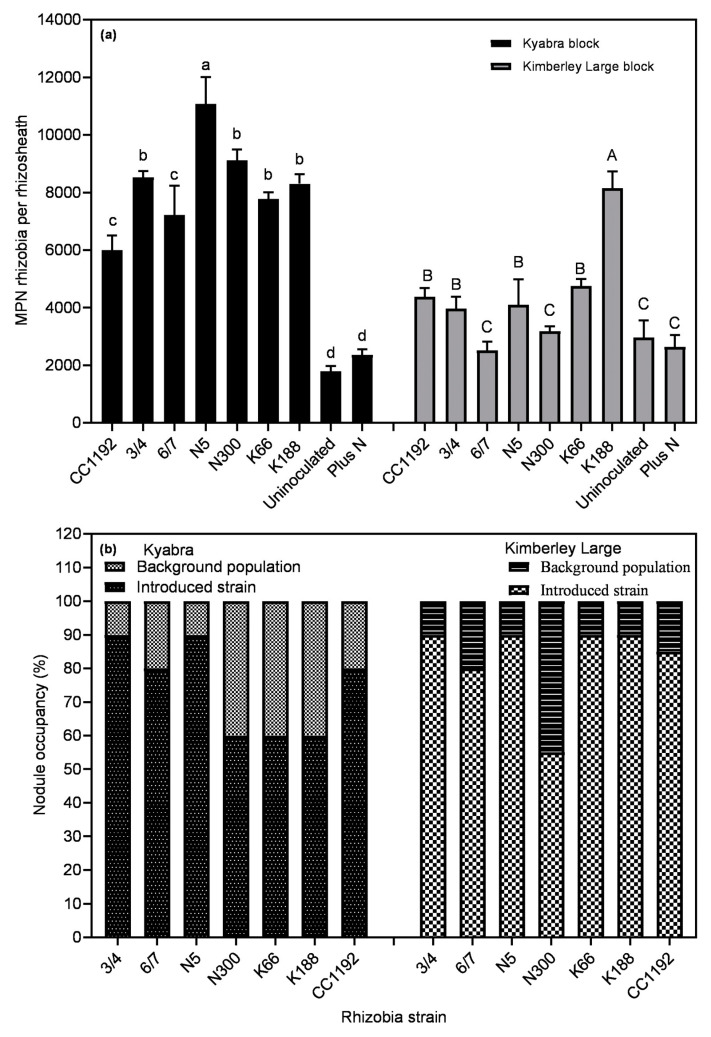
(**a**) MPN of rhizobial strains in the rhizospheres of Kyabra and Kimberley Large in Kununurra soil after nine weeks of plant growth. Treatments that share the same letter (lowercase for Kyabra block bars and uppercase for Kimberley Large) are not significantly different according to Dunnett’s multiple comparison test (*p* ≤ 0.05) levels, respectively. Data represent mean ± SEM (*n* = four plants per treatment). (**b**) Proportion of nodule occupied by inoculant strain compared to resident population in Kununurra field-grown chickpea. Data represent four plots per treatments, consisting of nine plants per plot and 20 nodules per plant.

**Table 1 plants-14-00809-t001:** Nitrogen fixation traits of new chickpea rhizobial strains.

			Kyabra							Kimberley Large			
Strain	NN	DNW	SDW	PN	SE%	N_2_ Fixation Rate	Strain	NN	DNW	SDW	PN	SE%	N_2_ Fixation Rate
K72	64 ± 7	0.091 ± 0.01	0.418 ± 0.03	25.00 ± 3.10 **	81.6	24.68 ± 1.24 **	K72	67 ± 4	0.094 ± 0.01	0.648 ± 0.02	29.77 ± 2.22 ***	99.1	38.66 ± 1.90
N5	56 ± 3	0.063 ± 0.02	0.454 ± 0.03	22.29 ± 0.91	98.7	51.61 ± 5.20	6/7	82 ± 3	0.116 ± 0.01	0.751 ± 0.01	28.09 ± 4.36 ***	114.8	38.01 ± 3.40
K188	75 ± 6	0.122 ± 0.03	0.527 ± 0.01	20.49 ± 1.16	102.9	33.06 ± 0.17	K66	56 ± 2	0.067 ± 0.01	0.453 ± 0.02	26.40 ± 3.13 **	69.3	48.19 ± 2.10
N300	72 ± 3	0.095 ± 0.01	0.449 ± 0.02	19.60 ± 2.31	87.7	40.07 ± 3.65	3/4	88 ± 2	0.094 ± 0.01	0.709 ± 0.02	22.76 ± 2.02 *	108.4	41.67 ± 4.20
6/7	78 ± 4	0.112 ± 0.02	0.443 ± 0.01	19.17 ± 3.23	86.5	38.10 ± 3.91	N5	64 ± 6	0.069 ± 0.03	0.715 ± 0.02	20.52 ± 2.01 **	109.3	53.03 ± 2.00
K66	44 ± 4	0.057 ± 0.01 *	0.311 ± 0.01	17.17 ± 1.40	60.7	46.65 ± 2.40	N300	58 ± 2	0.072 ± 0.01	0.536 ± 0.01	15.61 ± 2.21	82.0	52.65 ± 3.20
CC1192	66 ± 8	0.102 ± 0.01	0.463 ± 0.02	17.08 ± 1.52	90.4	39.49 ± 4.50	K203	66 ± 4	0.067 ± 0.02	0.511 ± 0.01	14.62 ± 1.10	78.1	36.38 ± 2.51
3/4	58 ± 7	0.083 ± 0.01	0.341 ± 0.05	14.27 ± 1.42	66.6	45.92 ± 5.94	CC1192	71 ± 3	0.083 ± 0.02	0.654 ± 0.02	14.48 ± 1.21	100.0	45.62 ± 3.70
K208	57 ± 5	0.087 ± 0.02	0.326 ± 0.02	14.03 ± 1.37	63.7	37.85 ± 3.21	K188	53 ± 5	0.065 ± 0.02	0.447 ± 0.01	13.60 ± 1.09	68.3	52.94 ± 4.60
K8	49 ± 4	0.082 ± 0.03	0.374 ± 0.04	14.00 ± 2.21	73.0	32.41 ± 2.11	K208	67 ± 5	0.076 ± 0.02	0.564 ± 0.01	13.12 ± 1.74	86.2	39.04 ± 3.60
K203	52 ± 6	0.097 ± 0.02	0.321 ± 0.03	12.80 ± 2.19	62.7	35.30 ± 3.81	K8	63 ± 2	0.072 ± 0.01	0.454 ± 0.02	12.40 ± 2.22	69.4	37.41 ± 1.11
Uninoc.			0.291 ± 0.30	7.94 ± 1.20			Uninoc.			0.247 ± 0.01	5.31 ± 2.20		
Plus N			0.512 ± 0.01	23.15 ± 2.04			Plus N			0.654 ± 0.03	26.91 ± 2.01		

Data for nodule number/plant (NN), nodule weight (g DNW), shoot weight (g SDW), plant nitrogen content (mg PN), symbiotic effectiveness (SE%) and N_2_ fixation rate (µmol H_2_ g^−^ DW nodule per plant—NFR) of different inoculation treatments and chickpea cultivars at 32 d after sowing. “Uninoc” stands for “uninoculated”. Data represented as mean ± SEM (*n* = 6 plants). Values with * are significantly different from CC1192, where *, ** and *** represent significance at 0.05, 0.001 and 0.0001 probability levels, respectively.

**Table 2 plants-14-00809-t002:** ANOVA of means for the effects of rhizobia inoculation on symbiotic traits of chickpea.

Sources of Variation	Nodule DW			Shoot DW		Shoot % N		RSE%	
	DF	% of Total Variation	Mean Square	% of Total Variation	Mean Square	% of Total Variation	Mean Square	% of Total Variation	Mean Square
Rs × Gp	8	13.1 ****	0.00	6.3 ***	43.60	9.5 ns	0.69	25.5 **	1378.00
Rs	8	58.5 ****	0.01	80.2 ****	4461.00	14.1 *	1.02	22.8 **	1233.00
Gp	1	24.9 ****	0.03	2.0 *	42.49	38.8 ****	22.37	0.2 ^ns^	73.02

Two-way ANOVA of means of symbiotic traits; shoot DW, nodule DW and RSE%. Here, Gp, Rs and Gp × Rs represent plant genotype, rhizobial strain and plant genotype × rhizobial strain interaction, respectively. *, **, *** and **** represent significance at 0.05, 0.001, 0.0001 and <0.0001 probability levels, respectively, while ns = no significance.

**Table 3 plants-14-00809-t003:** Relative symbiotic effectiveness of two chickpea genotypes inoculated with seven rhizobial strains.

Strain	Kyabra		Kimberley Large	
	RSE%	Rating	RSE%	Rating
CC1192	99.6	Effective	98.9	Effective
3/4	77.1	Poorly effective	102.0	Effective
6/7	78.6	Poorly effective	97.9	Effective
N5	78.3	Poorly effective	91.5	Effective
N300	82.7	Effective	87.1	Effective
K66	111.1	Effective	102.7	Effective
K188	97.7	Effective	83.1	Effective

Relative symbiotic effectiveness of Kyabra and Kimberley Large plants inoculated with seven chickpea rhizobial strains. Data represent means of four plots per treatment, consisting of nine plants per plot.

**Table 4 plants-14-00809-t004:** Final grain yield, 100 seed weight and seed N content of 2019 Kununurra field experiment.

Plant	Rhizobia	Yield (t/ha)	100 Seed Weight (g)	Seed N Content (%)
Kyabra	CC1192	2.90 ab	23.86	2.61
	3/4	2.60 abc	23.17	2.58
	6/7	2.60 abc	23.63	2.45
	N5	2.60 bc	23.15	2.65
	N300	2.70 abc	22.93	2.67
	K66	2.40 c	22.69	2.62
	K188	2.40 c	23.58	2.95
	−R/−N	2.40 c	22.52	2.83
	−R/+N	3.00 a	23.25	2.73
LSD, *p* = 0.05		0.25	0.89	0.02
Kimberley Large	CC1192	1.50 B	59.72	2.78
	3/4	1.70 A	60.41	2.39
	6/7	1.40 B	59.20	2.64
	N5	1.10 C	57.89	2.64
	N300	1.30 B	58.54	2.69
	K66	1.20 C	58.52	2.80
	K188	1.40 B	60.99	2.69
	−R/−N	1.30 B	59.40	2.64
	−R/+N	1.80 A	59.64	2.87
LSD, *p* = 0.05		0.52	3.08	0.01

Data are mean ± SD of Desi and Kabuli tonnes/ha and 100 seed weight for the 2019 Kununurra field trial *(n* = four plots consisting of nine plants per plot for seed %N data). LSD and *p* stand for least significant difference and probability level respectivly. Means followed by different letter (lowercase for Kyabra bars and uppercase for Kimberley Large) are significantly different, *p* < 0.05.

## Data Availability

Data associated with this study have not been deposited into a publicly available repository because our research is ongoing, and the funding body will decide when they should be made public. However, they could be available upon request.

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
