# Peer review of "Symbiotic Effectiveness, Rhizosphere Competence and Nodule Occupancy of Chickpea Root Nodule Bacteria from Soils in Kununurra Western Australia and Narrabri New South Wales Australia"

_plants, 2025, doi:10.3390/plants14050809_

Round 1
Reviewer 1 Report
Comments and Suggestions for Authors
The study screened 3200 rhizobia isolates from different regions of Australia, conducting a comprehensive assessment of their morphology, genetics, and symbiotic effectiveness. This large-scale screening and evaluation provided valuable data for discovering new, highly efficient rhizobia. The study not only conducted greenhouse experiments but also validated the performance of rhizobia under field conditions, ensuring the reliability and practical application value of the experimental results. The nitrogenase activity of rhizobia was determined through hydrogen gas release experiments (Hâ‚‚ evolution assay), providing direct evidence of the rhizobia’s nitrogen-fixing capability. The study evaluated the survival ability of rhizobia in the rhizosphere and their nodulation occupancy, revealing their competitiveness and adaptability in the field environment. The study identified new rhizobia strains (such as N5, N300, K72, and 6/7), which showed excellent performance in nitrogen fixation capability and adaptability, even surpassing existing commercial strains like CC1192.
Questions:
1) Although significant strain differences were found in greenhouse experiments, these differences were not pronounced in field experiments. This might be related to the complexity of the field environment, but the study did not delve into the reasons for this phenomenon.
2) The study was based on field experiment data from a single year, lacking support from data across multiple years or locations, which might not fully reflect the long-term performance and adaptability of the strains.
3) The study did not investigate the influence of other soil microbial communities on the symbiotic effectiveness and competitiveness of rhizobia, which could be an important confounding factor.
4)The study did not analyze the economic benefits of the new rhizobia strains, making it impossible to assess their practical application value in agricultural production.
5) Although the study conducted genetic diversity analysis through RAPD-PCR, it did not explore the molecular interaction mechanisms between rhizobia and host plants, especially gene expression and regulation related to nitrogen fixation.
6) Fig. 1 provides the details of the DNA marker.
Author Response
Author Response to Reviewer 1
Questions:
Reviewer comment #1: Although significant strain differences were found in greenhouse experiments, these differences were not pronounced in field experiments. This might be related to the complexity of the field environment, but the study did not delve into the reasons for this phenomenon.
Author Response #1: Thank you for mentioning this. We have touched on this on lines 506-517 in the discussion section of the revised manuscript.
Reviewer comment #2: The study was based on field experiment data from a single year, lacking support from data across multiple years or locations, which might not fully reflect the long-term performance and adaptability of the strains.
Author Response #2: Thank you for this comment and we agree with you that the strains need to be tested in multiple soils over several years. Based on Australia’s guidelines for selection, screening and recommendation of new inoculant strains, these strains are not yet at the stage for commercial release. Thus, we await future funding to continue the next phase of the project.
Reviewer comment #3: The study did not investigate the influence of other soil microbial communities on the symbiotic effectiveness and competitiveness of rhizobia, which could be an important confounding factor.
Author Response #3: Thank you for this comment. Prior to the field trail, we tested the soil for the presence of indigenous rhizobia to address the competitive effect of background rhizobia on infectivity and symbiotic effectiveness. Nodulation was not affected by other soil microorganisms. This information has been added to the revised document on lines 251-257 in the materials and method section. Also, we determined nodule occupancy and strain populations in the rhizosheath to examine the competitiveness of the inoculant strains against background rhizobial populations, with results shown in Figure 5 of the revised manuscript.
Author Response #4: The study did not analyze the economic benefits of the new rhizobia strains, making it impossible to assess their practical application value in agricultural production.
Author Response #4 Thank you for this comment. The initial funding at the commencement of this project was to collect, screen and identify chickpea rhizobial isolates that exhibited putative heat tolerance to the hash environmental conditions in the Kimberley region of Western Australia. Introducing new efficacious chickpea strains for symbiotic nitrogen fixation would reduce the need by farmers to use excessive amounts of inorganic N fertiliser in the cultivation and growth of Kyabra and Kimberley Large chickpea varieties. We have summarised the rationale in the introduction of the manuscript on lines 60-76.
Reviewer comment #5: Although the study conducted genetic diversity analysis through RAPD-PCR, it did not explore the molecular interaction mechanisms between rhizobia and host plants, especially gene expression and regulation related to nitrogen fixation.
Author Response #5: Thank you for this comment. We agree with you, but we were financially limited. Funding for molecular study of the strains was not provided, although it would be prudent to do the work. If funding becomes available, detailed molecular study of the strains will be the next phase of the project.
Reviewer comments #6: Fig. 1 provides the details of the DNA marker.
Author Response #5: Thank you for these comments. Details of the DNA marker has been provided in the legend of Figure 1 on lines 345-346 of the revised manuscript.

Reviewer 2 Report
Comments and Suggestions for Authors
This ms showed new rhizobial strains derived from naturalised soil populations exhibited better adaptation to local soil conditions than commercial chickpea inoculant strain
CC1192, which is interesting for the area. However, I have to reject the ms as I see that the ms is mostly descriptive, and there lacks of novelty and core thoughts about the topic. For example, the abstract, it is bit repetive,and no new ideas are provided.
Also, as I go through the ms, the information is quite messy, and needs more organized.
Nitrogenase Activity Determination assay is wield. Also,the pictures of fresh root nodule should also be included.
Author Response
Author Response to Reviewer 2
Reviewer Comment #1: This ms showed new rhizobial strains derived from naturalised soil populations exhibited better adaptation to local soil conditions than commercial chickpea inoculant strain CC1192, which is interesting for the area.
Author Response #1:
Thank you for your synopsis. We were excited to find and identify new rhizobia strains (such as N5, N300, K72, and 6/7), which showed excellent performance in nitrogen fixation capability and adaptability, even surpassing the existing commercial strain CC1192.
Reviewer Comment #2: However, I have to reject the ms as I see that the ms is mostly descriptive, and there lacks of novelty and core thoughts about the topic. For example, the abstract, it is bit repetive,and no new ideas are provided. Also, as I go through the ms, the information is quite messy, and needs more organized.
Authur Response #2:
Thank you for your comment. This project was to collect, screen and identify chickpea rhizobial isolates that exhibited putative heat tolerance to the hash environmental conditions in the Kimberley region of Western Australia. Introducing new efficacious chickpea strains for symbiotic nitrogen fixation will reduce the need by farmers to use excessive amounts of inorganic N fertiliser in the cultivation and growth of Kyabra and Kimberley Large chickpea varieties. The novelty and core thoughts are about finding ‘novel’ strains to improve the efficiency of symbiotic nitrogen fixation in chickpea in northern Australia using recognised and peer reviewed procedures.
The process of selection for new and more efficient strains follows protocols that are recognised as necessary to determine the veracity of results. Thus, in writing this manuscript, protocols, experimental design and procedures were descriptive, in that they enable a reader to repeat the experiment themselves if they so desire, using the detailed materials and methods. Strain selection protocols for elite rhizobial strains for specific host legumes has been in practice for over 100 years. Methodology has improved over time especially with the advances in molecular biology. We used these recognised methods for isolation, screening and authentication to characterise each strain BEFORE use in glasshouse or field trials to save costly time and resources.
You will find that the materials and methods are set out in sequence from the collection of seed, strain isolates and how they are characterised and tested. The corresponding results sections are labelled to match the method, so that a reader can read the method and find the related data in the results section under a matching heading.
Authur Response #3:
Nitrogenase Activity Determination assay is wield. Also,the pictures of fresh root nodule should also be included.
Authur Response #3:
Thank you for this comment. The nitrogenase activity was determined and used to express a nitrogen fixation rate. These results were reported in Table 1. Nitrogen fixation rate ranged from 32.4 (strain K8) to 51.6 (strain N5) µmol H2 g- DW nodule per plant. Strain N5 was 30% higher in NFR than the current commercial strain CC1192.
A traditional method of measuring N2-fixation effectiveness focuses on shoot biomass and seed yield assessment, even though 30-60 % of acquired N is partitioned to below ground tissues (Turpin et al. 2002). Thus, the aim of our choice of nitrogenase activity assay in the manuscript was to assess the efficiencies of symbiotic N2-fixation in Kyabra and Kimberley Large chickpea genotypes and rhizobial strain combinations to assist in the selection of rhizobial symbionts to improve chickpea production in northern Australia. This assay is non-invasive so it can be used to measure N2-fixation in intact plants, without disrupting the root system and it is a real-time monitoring which allows for continuous observation of N2-fixation activity under different environmental conditions.
It is worth noting that some previous researchers also used nitrogenase activity to measure N2-fixation in legumes including chickpea. Please see a few of publication citations below regarding this matter;
- Turpin JE, Herridge DF, Robertson MJ (2002) Nitrogen fixation and soil nitrate interactions in field-grown chickpea (Cicer arietinum) and fababean (Vicia faba). Australian Journal of Agricultural Research 53 (5):599-608. https://doi.org/10.1071/AR01136
- Dong, Z. and Layzell, D.B., 2002. Why do legume nodules evolve hydrogen gas. Nitrogen fixation: global perspectives. Wallingford, UK: CABI Publishing, pp.331-335.
- Golding, A.L. and Dong, Z., 2010. Hydrogen production by nitrogenase as a potential crop rotation benefit. Environmental Chemistry Letters, 8, pp.101-121.
- Tillard, P. and Drevon, J.J., 1988. Nodulation and nitrogenase activity of chickpea cultivar INRA 199 inoculated with different strains of Rhizobium ciceri. Agronomie, 8(5), pp.387-392.
- Sindhu, S.S., Dadarwal, K.R. and Dahiya, B.S., 1986. Hydrogen evolution and relative efficiency in chickpea (Cicer arietinum L.): effect of Rhizobium strain, host cultivar and temperature.
- Owaresat, J.K., Siam, M.A., Dey, D., Jabed, S., Badsha, F., Islam, M.R. and Kabir, M.S., 2023. Factors impacting rhizobium-legume symbiotic nitrogen fixation with the physiological and genetic responses to overcome the adverse conditions: A review. Agricultural Reviews, 44(1), pp.22-30.
Regarding pictures of fresh root nodule, we have included it in the revised manuscript. Please see Figure 3 on line 433.

Reviewer 3 Report
Comments and Suggestions for Authors
Dear Authors,
I can clearly see that this body of work has been building over several years and commend you for the rigour you have shown with regards to the numbers required for the analyses. Although I feel the paper would have benefitted from molecular identification of the strains from resulting nodules (in the glasshouse and field trials), I can understand why RAPD was employed (time and cost savings). However I would caution the Authors about making definitive statements regarding 'identity' based upon RAPD profiles. It would also help if the Authors could provide RAPD gels illustrating the differences between strains that were used as inoculants. For future use, I would suggest Sanger sequencing as these sequences could be made publicly available and could be used by the broader community.
I found following the Materials and Methods and Results slightly difficult; the subsections in these two sections do not match and I felt that the subsections in the Results flowed better. Could the Authors please attempt to rewrite the Materials and Methods to better fit the Results? Also it would help if the Authors were very clear about which results correspond to which analysis - the term 'glasshouse' for instance is never used in the Results. Please remember that the audience does not have such an intimate knowledge of your study design and strains as you do, rather provide more information - it easy to become lost between all the strains and parameters.
With regards to the MPN analysis, again I would caution the Authors against assigning it more discriminatory power than it has - I cannot see how the MPN approach could discriminate between an inoculant and indigenous/resident rhizobial strain, for instance. Surely there would be resident rhizobia capable of interacting with these chickpea varieties - I cannot think that specificity between rhizobium and host is so strict that the plant would filter out anything that is not an inoculant.

Author Response
Author Response to Reviewer 3
Reviewer comments #: I can clearly see that this body of work has been building over several years and commend you for the rigour you have shown with regards to the numbers required for the analyses. Although I feel the paper would have benefitted from molecular identification of the strains from resulting nodules (in the glasshouse and field trials), I can understand why RAPD was employed (time and cost savings). However I would caution the Authors about making definitive statements regarding 'identity' based upon RAPD profiles. It would also help if the Authors could provide RAPD gels illustrating the differences between strains that were used as inoculants. For future use, I would suggest Sanger sequencing as these sequences could be made publicly available and could be used by the broader community.
Author Response #1: Thank you for this comment and constructive caution. You are correct in assuming that we did not have time and funding for detailed molecular studies, and for RAPD gels that would illustrate the differences between strains that were used as inoculants. The results of that screening are not included in this manuscript as they were previously published in Oparah, I.A., Hartley, J.C., Deaker, R. et al. Symbiotic effectiveness, abiotic stress tolerance and phosphate solubilizing ability of new chickpea root-nodule bacteria from soils in Kununurra Western Australia and Narrabri New South Wales Australia. Plant Soil 495, 371–389 (2024). https://doi.org/10.1007/s11104-023-06331-w). See gel photo in attached file.
Reviewer comments #2: I found following the Materials and Methods and Results slightly difficult; the subsections in these two sections do not match and I felt that the subsections in the Results flowed better. Could the Authors please attempt to rewrite the Materials and Methods to better fit the Results? Also, it would help if the Authors were very clear about which results correspond to which analysis - the term 'glasshouse' for instance is never used in the Results. Please remember that the audience does not have such an intimate knowledge of your study design and strains as you do, rather provide more information - it easy to become lost between all the strains and parameters.
Author Response #2: Thank you for this comment. We have worded the heading of the sections in Materials and method to suit that of the results
Reviewer comments #3: With regards to the MPN analysis, again I would caution the Authors against assigning it more discriminatory power than it has - I cannot see how the MPN approach could discriminate between an inoculant and indigenous/resident rhizobial strain, for instance. Surely there would be resident rhizobia capable of interacting with these chickpea varieties - I cannot think that specificity between rhizobium and host is so strict that the plant would filter out anything that is not an inoculant.
Author Response #3: Thank you for this comments and constructive caution. We are aware of the limitations of MPN’s and the fiducial limits of estimates for accuracy. After isolating the contents of the nodules that were collected, we performed RAPD profiles on the isolates and then compared their fingerprints with those of the inoculant strains to determine percentage of introduced rhizobial populations against the background population. See Figure 5. Applying high numbers of inoculant rhizobia to the seed at sowing favours the host plant to form nodules with that strain as the seedling emerges. The inoculation process introduces the inoculant strain in very high numbers adjacent to the seedling root infection sites, thus outcompeting indigenous rhizobia that may be present in lower numbers in soil.

Round 2
Reviewer 2 Report
Comments and Suggestions for Authors
Thank you for the revision, as well as some reasonable arguments. The manuscript can be accepted now.
Author Response
Thank you very much for reviewing our manuscript.
Reviewer 3 Report
Comments and Suggestions for Authors
Dear Authors,
It would appear that you did not notice the additional document I had provided with suggestions for grammatical improvements. I have attached this again.
With regards to distinguishing between the inoculants on the basis of RAPD profiles for which you provided a gel photo with all of the test strains in your response to me - I could not tell the difference between the profiles for strains 6/7 and N5 nor between the profiles for N300 and K188. I was also suggesting that you add this gel photo to the paper (it could perhaps replace one of the other gel photos). Would it be possible that in some instances certain resident rhizobia could have the same RAPD profile as the inoculant strain and thereby the inoculant 'number' becomes inflated? I would suggest the manuscript should mention the possibility.

Author Response
Response to Reviewer #3
Dear Reviewer,
Thank you very much for taking your time to review our manuscript. Our apologies for not noticing the additional document you had provided with suggestions for grammatical improvements. We have responded and made the changes you suggested in the revised manuscript.
We have included a new gel photo. The gel was run at a longer time and shows some differences in 6/7 and N5 and between the profiles for N300 and K188 than the one we previously provided.
Yes, it is possible that in some instances certain resident rhizobia could have the same RAPD profile as the inoculant strain and thereby the inoculant 'number' becomes inflated. As you have suggested this has been mentioned in the manuscript on lines 615-617. Thank you
Responses to specific comments for Round 1.
Reviewer Comment #1: Lines 54 and 55: For ease of reading, I would propose changing this sentence to “With the latter becoming the commercial strain from 1977 [5].” Or anything in that vein, as the double ‘CC192’ is not easy on the eyes.
Author Response #1: Line 54-55: Sentence has been changed to “With the latter becoming the commercial strain from 1977”. Thank you
Reviewer Comment #2: Lines 81 to 84: This is a somewhat long and confusing sentence; it might the use of the terms “infective” and “ineffective” so close together. Might the meaning come across clearer if the Authors add “(i.e., ‘cheaters’ with a reference)” or “with that host” after “ineffective in fixing N2”. However, the Authors want to address it – this sentence just needs more clarity. I had to reread it a couple of times.
Author Response #2: Line 82: “with that host” and “in fixing N2” have been rearranged to clarify the sentence. Thank you
Reviewer Comment #3 : Line 96: Please change “gene” to the plural and subsequently “is” would need to become “are”.
Author Response #3: Line 96: “gene” has been changed to “genes” and “is” to “are”. Thank you
Reviewer Comment #4: Line 99: Again, for ease of reading I would propose the Authors change “for the Mesorhizobium species” to “of Mesorhizobium species”.
Author Response #4: Line 99: “for the Mesorhizobium species” has been changed to “of Mesorhizobium species”. Thank you
Reviewer Comment #5: Line 102: Add “the” before “recent diversity”.
Author Response #5: Line 102: “the” has been added before “recent diversity”. Thank you
Reviewer Comment #6: Line 104: I would propose to add “strains” after “Mesorhizobium”.
Author Response #6: Line 104: “strains” has been added after “Mesorhizobium”. Thank you
Reviewer Comment #7: Lines 107 and 108: For ease of reading perhaps consider changing “cultivars with capacity for production” to “cultivars with production capacity”.
Author Response #7: Line 107 and 108: “cultivars with capacity for production” has been changed to “cultivars with production capacity”. Thank you
Reviewer Comment #8: Line 113: Technically “rhizobia” is the plural and so I would suggest changing “legume” to the plural as well.
Author Response #8: Line 113: “legume “has been changed to “legumes”
Reviewer Comment #9: Line 159: The section “in Petri plates” could be removed.
Author Response #9: Line 160: “in Petri plates” has been removed from the sentence.
Reviewer Comment #10: Question: Could the Authors please indicate how the isolates are stored longterm (glycerol stock stored at – 80 °C, for example) and where or by which institute? This information would fit well after line 164.
Author Response #10: Lines 164-170: Long term storage of strains in glycerol stock at -80oC has been included in the paragraph. Thank you
Reviewer Comment #11: Question: Could the Authors please provide more information on how the strains were grown for extraction – how much of what broth was inoculated, was it shaken? At what temperature and at what speed for how long? Which DNA extraction protocol was used? Should this detail be provided in reference [14], then please make a statement to that effect.
Author Response #11: Line 177 “using protocols detailed by Oparah et al. [14]” has been added to address this issue. Thank you
Reviewer Comment #12: Comment: Lines 168 to 170 – Should the PCR cycle conditions, and agarose gel set-up be provided in reference [14]; then please point this out. Add a sentence to this effect.
Author Response #12: Line 177 “using protocols detailed by Oparah et al. [14]” has been added to address this issue. Thank you
Reviewer Comment #13: Line 170: I suspect there might be an additional space between “strains” and “[14]”.
Author Response #13: Line 180: Additional space between “strains” and “[14]” has been closed. Thank you
Reviewer Comment #14: Line 239: There is an “i” missing in the equation; it currently reads “fertlized”.
Author Response #14: Line 253: “fertlized” has been corrected to “fertilized. Thank you
Reviewer Comment #15: Line 253: Remove “the” from in front of “Cununurra”.
Author Response #15: Line 277: “the” has been removed from in front of “Cununurra”. Thank you
Reviewer Comment #16: Line 257: I would add “The” in front of “Seven”, of course changing the upper case “S” to a lower case. Starting the sentence with “Seven” creates the impression that it is a subset of strains (I even reread the above segment, to make sure that seven was the total number of strains included for field trial).
Author Response #16: Line 267: “The” has been put in front of Seven and “S” in seven has been replaced with ‘s”
Reviewer Comment #17: Line 286: If I am not mistaken this should refer to “equation (4) below” and not “equation (1) below”.
Author Response #17: Line 286: Yes, you are right. “Equation (1) has been corrected to “equation (4)”. Thank you
Reviewer Comment #18: Comment on the basis of Lines 287 to 303: I cannot see how this MPN approach would distinguish the applied inoculant rhizobial strains from those already present in the soil? Could the Authors please add a sentence as to how this approach would distinguish between the inoculant and any ‘background’ rhizobia before describing how it was performed?
Author Response #18: Line 327-300: A sentence has been added as to how MPN can distinguish between “inoculant strain” and “background “rhizobia”. Thank you
Reviewer Comment #19: Line 305: I suspect there is an additional space between “sowing” and “were”.
Author Response #19: Line 317: Additional space between “sowing" and “were “has been closed. Thank you
Reviewer Comment #20: Question on the basis of Lines 304 to 310 and Figure 1: How unique are RAPD fingerprints? Do the Authors have a gel with all seven of the strains used as inocula for the field trial? On the basis of the logic of this section – such a gel would be informative to show that it is at least possible to discriminate between these seven strains. At the moment the two gels included in Figure 1 do not show banding patterns for strains N300, K66 and K188. I am assuming that potential rhizobia were isolated from the resulting nodules, purified (so a single strain was used to generate the ‘field’ RAPD profile) and this was then compared to the RAPD profile of the pure culture used as inoculum? I think more detail is required for this analysis. Why did the Authors not make use of Sanger sequencing to confirm the identity of the strains from resulting nodules? These sequences could be made freely available on public databases, and I suspect it might be a more accurate identification tool as RAPD profiles.
Author Response #20: Line 343: A gel with the 7 inoculant strains on it has been added to the revised manuscript. We agree with you that Sanger sequencing to confirm the identity of the strains from resulting nodule would be better, but we were financially limited. Funding for molecular study of the strains was not provided, although it would be prudent to do the work. If funding becomes available, detailed molecular study of the strains will be the next phase of the project. Thank you
Reviewer Comment #21: Comment Lines 321 to 327: The basis of this study is only the 7 strains chosen as putative inoculants and not the 3200 strains that features in reference [14]. I think therefor only discuss here these 7 strains.
Author Response #21: Line 346-347: We have discussed only the 7 strains. Thank you
Reviewer Comment #22: Comment Lines 329 334: I found this section very confusing. I think it would help a lot if the Authors were very clear upon which strains were included for this manuscript and for which particular analysis. It would also be helpful if the. Authors would clarify how many times RAPD analysis were performed and on which strains. In this section for instance, it would appear that the Authors are providing the results for the RAPD analysis done to test for nodule occupancy? However, I was under the impression that F21 was not included in the most recent field trial – so why mention it wasn’t found in the field? It was included in the glasshouse analyses; however, I don’t think RAPD analysis was performed for nodules resulting from these experiments.
Author Response #22: RAPD analysis was done for nodule isolate isolates collected from both 2018 and 2019 field trials. The RAPD of the isolates from the 2018 trial was used to select new strains including the 7 under consideration in this manuscript while RAPD analysis of the 2019 trial nodule isolates was used for the “nodule occupancy” analysis. You are right, F21 was not included in the most recent field trial but the 2018 one and the comment made about it was regarding 2018, but we have taken it out to avoid confusion. Thank you
Reviewer Comment #23: Cautionary comment regarding Lines 330 to 332: The red lines could indicate. shared genetic material, however from my understanding of RAPD analysis discrimination is based on size differences and it provides no indication of DNA sequence similarity – just because two strands have the same sized band would not necessarily indicate that have a similar sequence. However, it does not mean they do not – which is why I propose just ‘softening’ the statement.
Author Response #23: Lines 342-343: The statement regarding red lines on gel bands has been relaxed. Thank you
Reviewer Comment #24: Overall comment regarding Materials and Methods and Results: It does not appear as if the subsections for these two sections match and in fact, I think the subsections for the Results section are more well thought-out. I propose the Authors reconsider the subsection layout of the Materials and Methods section accordingly. Although I would still state to which experiment the results refer –for instance the word ‘glasshouse’ does not appear in the Results section. I am assuming that everywhere F21 is mentioned refers to glasshouse results as I do not think the strain was tested in the field.
Author Response #24: Thank you for this comment. We have worded the heading of the sections in Materials and method to suit that of the results
Reviewer Comment #25: Figure 2: The bracket on the unit for the Shoot weight graph is not closed; i.e., the second bracket is missing.
Author Response #25: Figure 2: On the vertical axis, the bracket is closed after “g DW” and can be seen as “Shoot weight (g DW)/plant”.
Reviewer Comment #26: Line 364: Please double-check but it would appear that there is an additional space between the bracket and “showed”.
Author Response #26: Line 375: Yes, you are right. The double space has been corrected
Reviewer Comment #27: Line 365: Might it not make more sense to change the semi-colon (;) to a colon (:)?
Author Response #27: Line 376: The semi-colon (;) has been changed to a colon (:). Thank you.
Reviewer Comment #28: Line 366: It might read easier if “Table 1” appears in brackets.
Author Response #28: Line 378: “Table 1” has been bracketed. Thank you
Reviewer Comment #29: Line 374: There might be an additional space between the full stop and “Strain”.
Author Response #29: Line 385: Additional space between the full stop and “Strain” has been deleted.
Reviewer Comment #30: Question Lines 377 and 388: Should the symbiotic effectiveness values not be presented as percentages?
Author Response #30: Line 388: Yes, symbiotic effectiveness values should be presented as percentages. The correction has been made. Thank you
Reviewer Comment #31: Table 1: I would actually remove “Nitrogen fixation traits” in the table (as it also mentioned in the table heading). Would it be possible to better space “Kimberley large”? 3
Author Response #31: Table 1: “Nitrogen fixation traits” in the table has been removed and Kimberley Large is spaced when the page is in landscape. Thank you
Reviewer Comment #32: Line 412: Remove “and” in “minimal and to non-existent”.
Author Response #32: Line 423: “and” in “minimal and to non-existent” has been removed. Thank you
Reviewer Comment #33: Line 447: I would change “rhizobia strain” (in both instances) to “rhizobial strain”– like it was used in Lines 397 and 405.
Author Response #33: Line 463: “rhizobia strain” has been changed to “rhizobial strain”
Reviewer Comment #34: Question Table 4: There are no letters (to indicate significance) for the
Kimberley Large data.
Author Response #34: Table 4: Letters has been added on Yield values of Kimberley Large to indicate significance. Thank you
Reviewer Comment #35: Question Lines 459 to 460: I cannot see how the MPN approach candifferentiate between the inoculant strains and ‘background’ or ‘indigenous’ rhizobia that were already present in the field. With the same reasoning I cannot see how the MPN approach can show the dominance of a particular inoculant – as the MPN method does not provide any information on identity but only on number of strains.
Author Response #35: RAPD analysis of the 2019 trial nodules was used for the nodule occupancy analysis. We are aware of the limitations of MPN’s and the fiducial limits of estimates for accuracy. After isolating the contents of the nodules that were collected, we performed RAPD profiles on the isolates and then compared their fingerprints with those of the inoculant strains to determine percentage of introduced rhizobial populations against the background population. See Figure 5. Applying high numbers of inoculant rhizobia to the seed at sowing favours the host plant to form nodules with that strain as the seedling emerges. The inoculation process introduces the inoculant strain in very high numbers adjacent to the seedling root infection sites, thus outcompeting indigenous rhizobia that may be present in lower numbers in soil. Thank you
Reviewer Comment #36: Line 478: I would change the “p” in “proportion” to an uppercase.
Author Response #36: Line 494: the “p” in “proportion” has been changed to an uppercase. Thank you
Reviewer Comment #37: Lines 500 and 501: I would rewrite “many others faster in growth than CC1192” to “many others grow faster than CC192”.
Author Response #37: Line 527-528: “many others faster in growth than CC1192” has been rewritten as “many others grow faster than CC192”.
Reviewer Comment #38: Line 506: Rewrite “there existed fast and slow-growers” to “and that both fast and slow-growers exist”.
Author Response #38: Line 533: “there existed fast and slow-growers” has been rewritten to “and that both fast and slow-growers exist”. Thank you.
Reviewer Comment #39: Line 520: I would replace the second “and” with a comma.
Author Response #39: Line 547: The second "and” has been replaced by a comma
Reviewer Comment #40: Line 534: Double-check the space between “i.e.” and “inoculation”.
Author Response #40: Line 561: The extra space between “i.e.” and “inoculation” has been closed. Thank you.
Reviewer Comment #41: Line 536: Check spacing between “some” and “host”.
Author Response #41: Line 563: The extra space between “some” and “host” has been deleted. Thank you
Reviewer Comment #42: Line 544: Check spacing between “can” and “cause”. I would for ease of reading remove the second “and” in the sentence.
Author Response #42: Line 571: The extra space between “can” and “cause” has been deleted and the second “and” removed. Thank you.
Reviewer Comment #43: Line 545: The sentence would be easier to read without the “the” in front of “partner selection”.
Author Response #43: Line 572 “the” in front of “partner selection” has been deleted.
Reviewer Comment #44: Line 573: Please remove the “a” in front of “naturilised” as “rhizobia” is the plural form of ‘rhizobium’.
Author Response #44: Line 600: “a” in front of “naturilised” has been removed. Thank you.
Reviewer Comment #45: Lines 582 to 584: This reasoning is what is missing the Materials and Methods section. This single sentence paragraph however appears strange, I propose to add it as the final sentence to the previous paragraph.
Author Response #45: Line 608-610: Sentence has been added made the final sentence to the previous paragraph. Thank you.
Reviewer Comment #46: Line 586: Double check spacing in front of “Like”.
Author Response #46: Line 612: The space in front of “Like” has been deleted. Thank you
Reviewer Comment #47: Line 587: Change “selected new strains” to “newly selected strains”.
Author Response #4: Line 613: “selected new strains” has been changed to “newly selected strains”. Thank you.
Reviewer Comment #48: Line 589: Change “by the” to “of the”.
Author Response #48: Line 617: “by the” has been changed to “of the”.
Reviewer Comment #49: Question Lines 589 to 593: I find it difficult to follow the reasoning behind the ideas in this sentence – how would low nodule dry weights in uninoculated plots reflect rhizobial strain and host plant compatibility? Is it rather not just the result of the rhizobia not being able to ‘travel’ far enough in the given season with prevailing field conditions? Perhaps the Authors could expand – to link the ideas together better?
Author Response #49: We think it might be easier to delete the second half of the paragraph because we were basically saying the same thing as in the paragraph above it …”more competitive than the naturalised populations because of the high numbers applied of the inoculants at sowing”. Thus, the sentence has been deleted from the revised manuscript. Thank you
Reviewer Comment #50: Line 605: Again, check the spacing between “that” and “diversity”.
Author Response #50: Line 629: The extra spacing between “that” and “diversity” has been closed. Thank you

Round 3
Reviewer 3 Report
Comments and Suggestions for Authors
Dear Authors,
My final suggestion would be to provide a more descriptive figure legend to Figure 1 - describing in the legend what the reader needs to know to make sense of Gel 1 and Gel 2.
I have no further comments or suggestions.
Author Response
Reviewer comment #: My final suggestion would be to provide a more descriptive figure legend to Figure 1 - describing in the legend what the reader needs to know to make sense of Gel 1 and Gel 2.
Author response #1: Lines 358-361: A description of Gel 1 and 2 has been amended in the legend of Figure 1. Thank you
Reviewer comment #2: I have no further comments or suggestions.
Author response #2: Thank you for your diligent review and constructive suggestions
